# The Reciprocal Interplay between Infections and Inherited Metabolic Disorders

**DOI:** 10.3390/microorganisms11102545

**Published:** 2023-10-12

**Authors:** Albina Tummolo, Livio Melpignano

**Affiliations:** 1Department of Metabolic Diseases, Clinical Genetics and Diabetology, Giovanni XXIII Children Hospital, Azienda Ospedaliero-Universitaria Consorziale, 70126 Bari, Italy; 2Medical Direction, Giovanni XXIII Children Hospital, Azienda Ospedaliero-Universitaria Consorziale, 70126 Bari, Italy; livio.melpignano@policlinico.ba.it

**Keywords:** infection, inherited metabolic disorders, immune system, mitochondria, acute metabolic decompensation

## Abstract

Infections represent the main cause of acute metabolic derangements and/or the worsening of the clinical course of many inherited metabolic disorders (IMDs). The basic molecular mechanisms behind the role of infections in these conditions have not been completely clarified. This review points out the different mechanisms behind the relationship between IMDs and infections, providing an overview of this still-under-investigated area. Classically, infections have been considered as the consequence of a compromised immune system due to a biochemical defect of energy production. An adjunctive pathogenetic mechanism is related to a genetically altered protein-attached glycans composition, due to congenital glycosilation defects. In addition, a dietary regimen with a reduced intake of both micro- and macronutrients can potentially compromise the ability of the immune system to deal with an infection. There is recent pre-clinical evidence showing that during infections there may be a disruption of substrates of various metabolic pathways, leading to further cellular metabolic alteration. Therefore, infective agents may affect cellular metabolic pathways, by mediation or not of an altered immune system. The data reviewed here strongly suggest that the role of infections in many types of IMDs deserves greater attention for a better management of these disorders and a more focused therapeutic approach.

## 1. Introduction

Inherited metabolic disorders (IMD) are a group of genetic conditions characterized by the altered function of a metabolic pathway due to a defect in either an enzyme or transport protein [1]. According to the last International Classification of Inborn Metabolic Disorders (ICIMD), more than 1450 are so far recognized and categorized [2].

Some categories of IMD, such as urea cycle disorders, amino acid disorders, organic acidemias, disorders of carbohydrate metabolism, fatty acid oxidation disorders, and mitochondrial disorders may be grouped into the functional category of IMDs at risk of acute metabolic decompensation (AMD) [3].

Acute metabolic decompensation in IMDs is characterized by rapid-onset deterioration of metabolic status leading to life-threatening biochemical perturbations (e.g., hyperammonemia, metabolic acidosis, hyperlactacidemia) and death, if not properly and rapidly treated. Intercurrent infections are the most common trigger of AMD [4] due to the increased metabolic demand during infection, leading to a catabolic state [5].

Infections can also worsen the clinical course of IMDs not subjected to AMD, provoking chronic disruption of metabolic parameters and an unsatisfactory clinical outcome.

Although not systematically examined, all the above result in greater morbidity, prolonged hospital stays and a worsening outcome in a number of IMDs [6,7].

The reason behind this close relationship can be, first of all, attributed to the fact that infections represent a stimulus to the functioning of the immune system, which is a large user of energy, whose production and recycling is essential for its correct functioning [8,9]. Recent studies have shown that mitochondria play a crucial role in the regulation and proper functioning of the immune response to pathogenic noxae participating in both innate and adaptive immunoresponses [10,11,12].

Lymphocytes refer to a different energetic source according to their state: resting T cells mainly refer to oxidative phosphorylation (OXPHOS), whereas effector T cells prioritize aerobic glycolysis during anabolic metabolism [13,14]. Differently from T cells, B cells, after stimulation, upregulate both aerobic glycolysis and mitochondrial oxygen consumption by OXPHOS [15].

For these reasons, metabolic disorders of energy production potentially function as immunopathies predisposing to different kinds of infections [16,17].

However, the link between infections and IMDs may involve other pathogenetic mechanisms of different cellular pathways, with infective agents functioning as effect or cause of the metabolic derangement. This review aims at providing an overview on the available evidence of the main mechanisms behind this relationship.

## 2. Infections Directly Affect the Cellular Metabolic Pathway through Disruption of Intermediates 

### 2.1. Urea Cycle Disorders

Urea cycle disorders (UCDs) are a group of genetic disorders caused by a loss of function in one of the enzymes responsible for ureagenesis [18], which are commonly distinguished in proximal (mitochondrial) and distal (cytosolic) disorders. The incidence of these disorders, estimated at around 1:35,000, is increasing over time as some UCDs are part of newborn-screening programs [19]. In both proximal and distal UCDs, infection may cause acute severe hyperammonemia, associated with increased morbidity and mortality [20]. Many reports describe the role of infections and sepsis in acute hyperammonemic episodes, highlighting that infection places a greater burden on the urea cycle, as protein breakdown is accelerated, leading to a catabolic state [21]. The relationship between infections and hyperammonemia has been studied by McGuire et al. [22] in a mouse model (spf-ash) of acute metabolic decompensation due to PR8 influenza virus infection in Ornitine Transcarbamilase Deficiency (OTC-D), the most common UCD, with an incidence ranging from 1:14,000 to 1:77,000 live births [23]

Both wild type (WT) and spf-ash mice showed reduced activity in the first two enzymes of the urea cycle—carbamoyl phosphate synthetase 1 (CPS1) and OTC—pointing out that a reduction of CPS1 and OTC enzyme activity are part of the hepatic physiology of PR8 infection, which might not be tolerated by a compromised urea cycle in spf-ash mice. In addition, differently from WT mice, spf-ash mice showed hyperammonia, probably secondary to reductions in the intermediates aspartate, ornithine and arginine.

These data lead to the suggestion that acute metabolic decompensation during infection may represent a failure to adapt to normal physiologic mechanisms due to an altered urea cycle.

### 2.2. Fatty Acids Oxidation Disorders

Fatty acids oxidation disorders (FAOD) are a heterogeneous group of IMDs secondary to the defective transport or β-oxidation of fatty acids, which are particularly involved in energy production during fasting and stress episodes [24,25]. 

The most severe forms are due to defective degradation of long-chain acylCoA dehydrogenase (LCAD), long-chain hydroxyacylCoA dehydrogenase (LCHAD) and trifunctional protein (TFP), with decompensation episodes characterized by hypoglicemia, metabolic acidosis, rhabdomyolysis and severe hepatopathy and cardiomyopathy [26].

Infections are a well-known trigger for FAOD metabolic decompensation. The best-known explanation is that memory CD8 T cells, lymphocytes with the ability to kill virus infected cells [27], characterized by the use of FAO as a source of energy in the activation state [14], undergo an altered capacity to face the infective agent.

Furthermore, CD8 T cells have also been shown to upregulate Carnitine Palmitoyl Transferase 1A (CPT1A), a mitochondrial membrane protein regulating long-chain fatty acid transport across the outer mitochondrial membrane, in the activation state [28]. Subjects carrying a mutant variant of CPT1A demonstrated an increased susceptibility to respiratory tract infection [29].

However, patients with FAOD receive an additional metabolic insult during infection, which is not mediated by an altered CD8 + function. Tarasenko et al. [30] realized a mouse model of metabolic decompensation by infecting mice with VLCAD (Acadvl^−/−^) and WT mice with PR8 influenza virus, aiming to determine a viral pneumonia. Acadvl^−/−^ mice showed lower blood glucose levels compared to WT, confirming an altered availability of FAO during infection in Acadvl^−/−^, due to an altered glucose homeostasis. Acadvl^−/−^ mice also showed specific perturbations in plasma acylcarnitine profiles during infection, with long-chain metabolites (C16–C18) increased in infected Acadvl^−/−^ mice and not in WT mice. 

Besides perturbations in long-chain fatty acids, the authors also found changes in metabolites involved in alternative metabolic pathways to compensate for FAO deficiency during metabolic decompensation due to infection. In particular, they found defects at numerous steps in long-chain FAO, including the carnitine cycle, acyl-CoA dehydrogenases, and the electron transport flavoprotein with an increased concentration of alternative substrates via residual fatty acid oxidation. In addition, increased markers for ketogenesis in the muscles and liver, secondary to the consumption of medium-chain triglycerides, and the activation of medium-chain FAO, were found. Although the adaptations found were tissue-specific and not well-represented in plasma acylcarnitines, the reported data help to better define the pathogenetic cascade associated with this viral infection in the FAO metabolism.

## 3. Diet Regimen Contributes to Alter Immune Response by Reducing Micro- and Macronutrients’ Availability

Many intermediary metabolism disorders are primarily and essentially treated with a nutritional approach, aimed at reducing concentrations of toxic substrates by reducing the assumption of nutrients that produce toxic products, or by bypassing the deficient enzyme providing more distal metabolites with the diet [31].

Phenylketonuria (PKU) is the most common inherited amino acid disorder, due to a deficiency of Phenylalanine Hydroxylase (PAH), which converts Phenylalanine (Phe) into Tyrosine (Tyr). To control blood Phe levels, a low-protein-regimen diet must be undertaken from birth by the affected subject [32].

There are a few studies on PKU and its association with infections; the ones available have highlighted changes in humoral immunity, but data include a limited number of patients. A restricted protein intake, which is also associated with an altered dietary fat intake, trace mineral and vitamin status, has demonstrated a capacity to influence humoral and cellular immune functions [33]. Decreased immunoglobulin levels have been reported in PKU patients by Passwell et al. [34], but the reasons behind this observation were not provided.

In experimental animal models, it has been shown that Phe decreased both antibody production and homograft rejection [35]. However, a direct influence of elevated plasma Phe levels on immune functions has not been reported yet, and in clinical practice, PKU patients are rarely admitted to hospital due to infections compared to other IMDs patients. In a study by Gropper et al. [36], mean plasma IgG and IgA concentrations in PKU children were significantly lower than values from similarly aged children. Moreover, positive correlations were obtained between plasma albumin and percentages and numbers of CD3+ and CD4+, between plasma IgG and interleukins 1 and 2, and between intakes of energy, protein, iron and plasma IgG levels, confirming that the protein omeostasis is linked to a normal functioning of the immune system.

Similar findings can be derived from a more recent study by Karazoz [37] in which the mean serum zinc (Zn) and iron levels of all patients were lower than the referral values of healthy children. Zn deficiency affects cells involved in both innate and adaptive immunity at the survival, proliferation and maturation levels. These cells include monocytes, polymorphonuclear, natural killer, T and B cells. T cell functions and the balance between the different T helper cell subsets are particularly susceptible to changes in Zn status [38].

Many studies have reported lower than normal micronutrients levels in PKU subjects [39,40,41,42]; therefore, regular nutritional status evaluation is important to identify possible nutrient deficiency and prevent secondary effects, including suboptimal immune-system functioning.

## 4. The Altered Metabolic Pathways Expressed in Lymphocytes May Affect the Immune Response to Infections

An altered immune response to infection may account for two different mechanisms, underlying corresponding IMDs: altered energy production and altered ligand–receptor interaction.

### 4.1. Altered Energy Production

#### 4.1.1. Mitochondrial Diseases

Mitochondrial diseases (MDs) are a large group of heterogeneous disorders resulting from mutations in either mitochondrial DNA (mtDNA) or nuclear DNA (nDNA), causing an altered OXPHOS and mitochondrial survival. The clinical effects of MDs are potentially multisystemic, involving organs with large energy requirements such as the heart, skeletal muscle, brain and immune system [43,44].

According to recent evidence, there are profound links between immune system and mitochondrial function. In particular, mitochondria play a pivotal role both in the innate and adaptive immune systems [45,46] and susceptibility to infections due to immune dysfunction is increasingly recognized in mitochondrial disorders, even though immunophenotypes have not been always described.

In the innate immune response to infection, particularly by viruses, mitochondria participate via linkages to effectors of pattern-recognition receptor (PRR) signaling [47], a class of receptors that can directly recognize the specific molecular structures on the surface of pathogens These are able to activate a signal cascade that ultimately results in the immune response. Regarding the adaptive immune response, different immune cell subtypes use distinct metabolic pathways to produce energy, according to their state.

Despite studies documenting the effects of infection in MD patients, immune dysfunction remains an underdiagnosed phenotype in MDs.

Evidence that the immune system is altered by an energetic defect come from multiple reports where MD patients have similar presentations to patients with primary immunodeficiencies, including unusual infections not commonly diagnosed in immunocompetent people [48]. Leukopenia has been reported in different MDs, including Barth syndrome, Pearson syndrome, Leigh syndrome, and other nonsyndromic forms of MD [49,50,51,52,53,54,55,56]. Recurrent hypogammaglobulinemia and reductions in natural killer, total CD8 T cells, and CD8 memory T cells were also noted [57]. These findings are most likely due to the depletion of mtDNA encoded components of OXPHOS which are critical for the immune cell function of the above cells. Functional and clinical characteristics of immune response alterations in the most frequently involved MD disorders are reported in Table 1.

In a cohort of 221 pediatric patients with mitochondrial disease, the global mortality rate was 14%; the two most common causes of death were sepsis (55%) and pneumonia (29%) [58]. In a retrospective study of 92 patients with mitochondrial neurogastrointestinal encephalomyopathy (MNGIE), 7.6% were noted to have a history of infections that provoked a worsening of symptoms [59]. The most common finding among 62 children (mean age = 7.4 years) affected by different types of MDs was recurrent or severe infections (89%), with contemporaneous symptoms of upper and lower respiratory tract infections [60].

Overall, the clinical findings of immunodeficiency in MD suggest that reductions in components of OXPHOS are critical for immune cell function and homeostatic maintenance. Immunoglobulin replacement therapy for clinical immunodeficiency during MD metabolic decompensation has been documented and can represent a successful approach in infective situations.

#### 4.1.2. Organic Acidemias

Organic acidemias (OAs) are due to a defect in intermediary metabolic pathways of carbohydrate, amino acids and fatty acid oxidation, which lead to an accumulation of organic acids in tissues and in urine [61].

Patients with OAs suffer from recurrent infections, which may cause a high morbidity and mortality rate. Some researchers have reported recurrent or unusual bacterial and/or viral infections, such as recurrent multiple molluscum lesions [62] and ecthyma gangrenosum due to a systemic Pseudomonas aeruginosa infection. In this last case, a malnutrition state, secondary to a protein-restricted diet, has been indicated as the main cause of the reported serious infection [63].

Although neutropenia has been reported in multiple studies, other components of the immune system are involved in the pathogenetic mechanism of the altered immune response in OA. A global effect of the disease on T and B lymphocytes has been reported, leading to adaptive immune defects that make them susceptible to infections [64,65]. B cell immunodeficiency and subsequent alterations of immunoglobulin levels have also been reported in cobalamin deficiency and in propionic acidemia [64].

#### 4.1.3. Glycogen Storage Diseases

Glycogen storage diseases (GSDs) are a group of rare conditions secondary to a genetically determined enzymatic defect of the metabolism of glycogen, so that it cannot be used for energetic purposes or build up properly in the liver [66].

To date, over 12 types of GSD have been identified and classified based on the enzyme deficiency and the affected tissue. Type I is the most common GSD and involves the liver, kidneys and intestine in subtype Ia, and also leukocytes in subtype Ib [67].

GSD1b is caused by a deficiency of glucose-6-phosphate translocase (G6PT), an enzyme involved in the last step of both gluconeogenesis and glycogenolysis [68]. In addition to the risk of hypoglicemia, hyperuricemia and hypertrigliceridemia, GSD1b also manifests with inflammatory bowel disease, recurrent infections, and persistent or intermittent neutropenia, which requires treatment with G-CSF [69].

The underlying mechanisms of neutrophil dysfunction are not well understood, but several hypotheses have been proposed [70] to explain why patients have frequent bacterial infections.

In a recent study by Jun et al. [71], the underlying cause of GSD-Ib related neutropenia was an enhanced neutrophil apoptosis. However, neutrophils from GSD-Ib patients also manifest functional dysfunction deriving from impairments in neutrophil glucose 6 phosphate (G6P) metabolism. G6PT interacts with the enzyme glucose-6-phosphatase-β (G6Pase-β) to regulate the availability of G6P/glucose in neutrophils during fasting. The altered G6PT interferes with the activity of the G6Pase-β/G6PT complex in neutrophils, impairing both their energy homeostasis and function and resulting in decreased glucose uptake and reduced neutrophil respiratory burst.

Neutrophil dysfunction in GSD-Ib have also been recently associated with an accumulation of 1,5-anhydroglucitol-6-phosphate (1,5AG6P) that lowers the phosphorylation of glucose, thus depressing the glycolytic pathway, essential for the immunometabolic activation [72]. In this context, empagliflozin, an inhibitor of the kidney sodium glucose cotransporter 2, also lowers serum 1,5AG6P in GSD-Ib patients. Preliminary data on a few G6PT deficient patients have shown that empagliflozin improves neutrophil count and function [73] and represents a promising therapeutic option for controlling neutrophil dysfunction in GSD1b patients.

### 4.2. Altered Ligand-Receptor Interaction

#### Congenital Disorders of Glycosilation

Glycosylation is a metabolic process essential for the proper functioning of a broad spectrum of proteins and lipids. Defects in genes encoding the formation of sugar nucleotides, or different steps of the glycosylation processes, result in the disruption of several glycosylation pathways and might lead to congenital disorders of glycosylation (CDGs). The role of glycans in the immune response can be recognized in two interrelated scenarios: the interaction between pathogens with host glycans for infection; the glycan composition of the immune cells’ surface influences the subsequent signaling pathways, and, consequently, the triggered immune response [74].

Immunological involvement is present in a subgroup of CDGs among which, those with major immunological involvement are ALG12-CDG, MOGS-CDG, SLC35C1-CDG and PGM3-CDG (Table 1), all characterized by different immunological dysfunction. The spectrum of immunological alterations may range from oral candidiasis, with an otherwise normal immunological function, to severe immunodeficiency with lethal infections [75,76].

Asparagine-Linked Glycosylation 12 (ALG12)-CDG patients can present with recurrent and severe infections. The immune hallmarks of the disease are the low serum IgG IgM, IgA levels [77]. Mannosyl-Oligosaccharide Glycosidase (MOGS) is the first enzyme involved in the processing of N-linked oligosaccharides. MOGS-CDG is a paradoxical case of immunological dysfunction: while being associated with an immunodeficiency phenotype (Table 1), MOGS-CDG patients present an increased resistance to viruses with glycosylated envelopes [78].

Mutations in SLC35C cause leukocyte adhesion deficiency type II (LAD II), leading to the defective transport of GDP-fucose from the cytoplasm to the Golgi lumen, where it is used as a substrate for fucosilation. Impaired and/or absent fucosilation negatively impacts the biosynthesis and function of selectin ligands and of various fucosilated proteins [79].

Phosphoglucomutase 3 (PGM3)-CDG is a congenital disorder of glycosilation associated with immunodeficiency and consequent recurrent bacterial and fungal infections, often characterized by increased levels of IgE [80]. Stray–Petersen et al. [74] identified three unrelated children with deleterious mutations in PGM3 who presented with recurrent infections, congenital leukopenia: neutropenia, B and T cell lymphopenia, and progression to bone marrow failure. The clinical phenotype was completed by skeletal dysplasia, dysmorphic facial features and cognitive impairment.

**Table 1 microorganisms-11-02545-t001:** Immune response alterations in mitochondrial and CDG disorders.

Type of Disorders	Cellular Alteration	Biochemical Alteration	Type of Infection	Reference
*Mitochondrial disorders*				
DNA depleting syndromes	decreased natural killer and CD8 T cells	hypogammaglobulinemia	pulmonary infections	[50,51]
Barth Syndrome	persistent or intermittent neutropenia	N/A	invasive aspergillosis, cutaneous zygomycosis	[53]
Leigh Syndrome	decreased B cells and memory T cells	N/A	viral airway infections, RSV bronchiolitis, otitis media, sepsis	[54]
MELAS (mitochondrial encephalomyopathy, lactic acidosis, stroke-like episodes) Syndrome	T cells altered function	N/A	COVID-19 infection	[55,56]
*Congenital glycosilation defects*				
ALG12 (Asparagine-Linked Glycosylation 12)-CDG	decreased B cells	decreased IgG, IgM, IgA	pneumonia, otitis media, ear/nose infections, sepsis	[77]
MOGS (Mannosyl-Oligosaccharide Glycosidase)-CDG	B and T cells, lymphocytic proliferation, neutropenia	decreased IgA, IgM, IgG	repeated sepsis by *E. coli*	[78]
SLC35C1-CDG	decreased B and T cell, neutrophilia, low neutrophilic mobility	N/A	respiratory infections mild to severe periodontitis, severe and/or localized cellulitis, gastroenteritis, recurrent sepsis	[79]
PGM3 (Phosphoglucomutase 3)-CDG	lymphopenia, with reverted CD4/CD8 ratio, eosinophilia congenital neutropenia, normal NK cells	high IgE level	respiratory tract, skin and oral infections, cutaneous abscesses by bacteria, viruses and mycetes	[80]

N/A: not available.

Targeted therapies to restore immune defects are only available for PGM3-CDG and SLC35C1-CDG [79,80]. Intravenous immunoglobulin administration has resulted in clinical improvement in many CDG patients. In general, a good response to antibiotic therapy leukocyte transfusion, steroids and G-CSF has been reported [81,82]. In severe immunodeficiencies, hematopoietic stem cell transplantation (HSCT) has been performed [83].

## 5. Discussion

The coexistence of infection and IMDs represents a reality in clinical practice and a well-known issue for all those involved in the management of these diseases [84,85]. The set of factors that underlie this relationship, however, has only been partially studied. Figure 1 schematizes the main passages of the interplay between infections and hereditary metabolic diseases.

Immunometabolism, which defines the role of the intermediary metabolism in immune cell function, is increasingly considered in the scientific literature and studies on mitochondrial diseases to demonstrate how an altered energy production may be the cause of reduced activity in the cells of innate and acquired immunity, predisposing to infections [10,11,12,13,14,15,16,17]. At the same time, some CDG are clearly associated with an altered immunological phenotype due to abnormal glycosilation and are rightly considered immunopathies [76].

The role that the dietary therapy of intermediate metabolism disorders can play in nutritional status and subsequently on the immune system, is even more poorly reported, but represents an issue to take into account when managing these diet regimens. A low zinc and iron intake is common to protein-restricted diet regimens not only in cases of PKU [86], and may potentially determine a low immune cell function. Furthermore, the frequent observation of overweight and metabolic syndrome in PKU adult patients [87,88], might predispose to adjunctive mechanisms of an altered immune system linked to multiple cytokines production and inflammatory process-activation [89] related to insulin resistance and diabetes mellitus type 2 [90], which however, needs to be substantiated with further evidence. The scarce availability of substrates for the urea cycle and the alteration of the concentrations of acylcarnitines in FAODs, come from preclinical studies and demonstrates how infection, particularly viral infections (and the resulting inflammatory cascade), may further alter the already-disrupted metabolic pathway and activate accessory metabolic pathways often responsible for further cellular damage. These findings suggest that supplementation with urea cycle intermediates, such as arginine or citrulline, already used in clinical practice, are particularly beneficial for UCD patients during infection and in long-chain FAOD; promoting short-to-medium-chain acylCoA beta oxidation can improve the pathogenetic cascade associated with viral infection [91,92].

An active area of research in the recent years is related to the study of gut microbioma and to its influence on human health. This relationship can develop via different mechanisms, including a close link with the immune system [93,94]. Gut microbiota, by metabolizing a wide range of metabolic products, can influence both the innate and adaptive immune systems, selecting and adjusting responses in the most appropriate manner [95]. A growing number of studies report the composition of gut microbiota in IEMs, in particular in PKU [96,97] and GSD [98,99], highlighting peculiar changes in its composition both related to diet treatment and the underlying disease [100]. The possibility of performing microbioma-based interventions in IMDs to modulate the immune system is a challenging area of research and could represent an adjunctive therapeutic strategy for IMDs.

Despite infections significantly affecting the course of a number of IMDs, many critical questions remain unanswered, and infection-based interventions, particularly the targeted ones, still experience significant limitations in clinical practice. The current treatment approaches toward immunological dysfunction are mainly limited to Ig administration for hypogammaglobulinemia, G-CSF for neutropenia or the use of wide-range antibiotics for recurrent infections. The only available targeted therapies directed to the immunological dysfunction in CDG are fucose supplementation in SLC35C1-CDG and HSCT in PGM3-CDG patients [76]. In mitochondrial diseases, some molecules able to promote mitochondrial fusion, such as resveratrol, have been demonstrated to improve adoptive cellular immunotherapy in several diseases [101], by increasing respiratory complex efficiency, on which memory T cell populations rely. However, most of this evidence comes from preclinical settings [102,103].

The clinical studies and data included here, although including preclinical studies, with weak evidence for clinical practice, provide an assessment of current evidence suggesting that infection processes mediated or not by immune dysregulation, play a crucial role in the course of many IMDs. Further studies are needed to elucidate the precise mechanisms linking infective agents and IMDs, which may lead to more targeted intervention strategies and to a more regular evaluation of the immunophenotype in IMDs patients at follow-up.

## 6. Conclusions

Given the failure of therapeutic approaches to a number of IMDs such as MDs and CGDs, and the difficulty in managing FAOD and UCD acute decompensations, new and adjunctive approaches should be explored. The data reviewed here strongly suggest that the role of infections in many types of IMDs deserves greater attention for a better management of these disorders during infection, towards the ultimate goal of improving outcomes and reducing mortality.

## Figures and Tables

**Figure 1 microorganisms-11-02545-f001:**
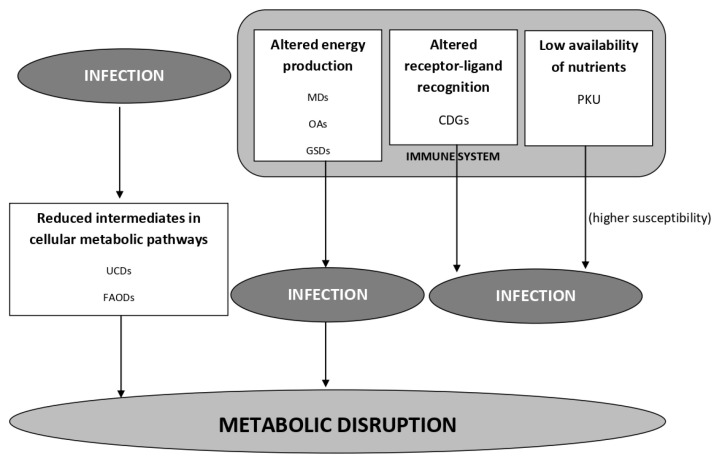
The reciprocal interplay between infections and hereditary metabolic diseases—The metabolic alteration acting on the proper functioning of the immune system can be the cause of the infection and, in the case of IMDs at risk of AMD, also the cause of metabolic disruption. On the other hand, infection can directly affect the cellular metabolism, resulting in a perturbation of substrates, leading to a greater risk of metabolic disruption. *Legend*: MDs (mitochondrial diseases); OAs (organic acidurias); GSDs (glycogen storage diseases); CDGs (congenital glycosilation disorders); PKU (Phenylketonuria); UCDs (urea cycle disorders); FAODs (fatty acids oxidation disorders).

## Data Availability

No data were created and/or analyzed for the purpose of this review. Therefore, a data availability statement is not applicable.

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
