# Peer review of "The Reciprocal Interplay between Infections and Inherited Metabolic Disorders"

_microorganisms, 2023, doi:10.3390/microorganisms11102545_

Round 1

Reviewer 1 Report

Dear Authors

Thank you for your paper. I have read it on two occassions and I struggled with its content and overall presentation  I found the flow of the paper difficult it was lots of short paragraphs and the cohesion and flow to me was stilted. I think more references are needed and I am not sure if the title matches the content I will try to highlight these difficulties

1. Overall I do not know if you fully met the brief of you title. In the paper you discuss infection and related this to not just immunity -You mention cellular pathways but this is not evidenced fully in the paper

For example the paragraph on urea cycle defects

1. A large section is given to explaining UCD this could be much shorter as this is not the focus of this paper

2. In UCD infection leads to high ammonia,  this is commonly seen in patients with UCD. In the mouse model no evidence is given if these were on sodium benzoate/ phenylbutyrate usual in human subjects to help the pathway function.

3 In UCD  ureagensis is always compromised and infection places a greater burden on the cycle as protein breakdown is acclerated

4. The work by Tarasenko is not explained in any detail how do amino acid disturbances in UCD due to infection extend beyond the liver? What about more evidence

5 Fatty acid oxidation MCAD is not a severe form of FAO- of all the FAO defects this is the one only needing treatment in illness and can be treated by a normal diet and an emergency regimen in illness. After the age of 12m a 12 h fast is recommended- unlike the other FAO which commonly need overnight feeding and TFP is known to have poor immuninology.

6. CD8T cells for the reader it would be good to have some background about these cells e.g lympocytes with the ability to kill virus infected cells. I would argue that in any IMD (FAO/ UCD) any stress due to an infection leads to abnormal metabolities as the pathway is compromised. My struggle was how does this relate to the title not just immune function?

7. Diet and altered immune function there is no greater explanation on the sentence 'indeed an increased susceptibility to infections due to  decreased immunoglobulin levels has been reported in PKU patients but this is only referenced by one paper and no explanation given to how. Additionally PKU subjects are rarely admitted to hospital with infections compared to UCD/FAO this needs to be addressed

8. Groppers paper is valid but what about more recent findings? I think the findings on Zn and iron could be challenged this is just one reference

9. Could it be argued that in mitrochondrial disorders immune resonses are chaged due to the defect in the mitrochondria, central to energy production? Which is the interplay between infections and IMD

10 Table 1 need the names of ALG 12-CDG/ MOGS

11 Glycogen storage disease - what about the new findings Empaglifozin

12 Arginine/ citrulline are supplemented in most UCDs

13 In OA mention unusual bacterai but what type and what it the mechanism of action

14 What about gut microbiota and immunity?

Other findings

·      Line 37 typo and death if not properly and rapidly treated

·      Line 40 repeat word acute

·      Lack of references for lines 46 to 50 only one reference. Ref for energy use for immune systems, only one reference for metabolic disorders and immune link

·      Line 53 spelling Lymophcytes

·      Line 57 to 59 spelling predisposing not predisponing

·      Spelling OTC  ornithine transcarbamylase deficiency most common but no reference to how common

·      Line 100 deputated? FAO energy is produced continually for movement not just in resting or stress

Line 142 is difficult to read hydoxylase sentence structure needs re wording

·      Line 143 English  since birth should be from birth

·      Line 162 ? spelling and meaning subtend corrispective ?

·      Line 178 does not make sense

I think more work is needed on this paper for it to fully match its title, more references and evidence and arguments presented. There is a lot of work gone into the manuscript but in its current form I do not think it fully answers the question you have proposed.

English needs to be revised in some parts

I thought the style of the paper was at times stilted- there are lots of short paragraphs and I think this alters the flow of the paper

Author Response

Thank you for your paper. I have read it on two occassions and I struggled with its content and overall presentation  I found the flow of the paper difficult it was lots of short paragraphs and the cohesion and flow to me was stilted. I think more references are needed and I am not sure if the title matches the content I will try to highlight these difficulties

R.Dear reviewer, thank you for your comment. We have added new references to the paper, extended the content and tried to make more consecutive the content, also by answering to the reviewers comments. Unfortunately we were not able to reduce the number of the paragraphs, because each of them is related to different inherited metabolic conditions, with different pathogenetic mechanisms explaining their relationship with infections.

Overall I do not know if you fully met the brief of you title. In the paper you discuss infection and related this to not just immunity -You mention cellular pathways but this is not evidenced fully in the paper

R.By this review we wanted to highlight an underinvestigated topic such as the relationship between IMDs and infections, which follows different pathogenetic mechanisms. The evidences are so far scarce, and rarely coming from clinical experience. We have decided to keep more generic the title of the paper, by eliminating its second part.

For example the paragraph on urea cycle defects:

A large section is given to explaining UCD this could be much shorter as this is not the focus of this paper.

R.Many thanks for this comment, which allow us to better focus the topic of the paper. We have now reduced the lengh of the paragraph, trying at the same time, to keep the core of its message and to include the response to other comments.

In UCD infection leads to high ammonia,  this is commonly seen in patients with UCD. In the mouse model no evidence is given if these were on sodium benzoate/ phenylbutyrate usual in human subjects to help the pathway function.

R.This comment gives us the possibility to clirify better : the experiments of McGuire et al,were performed on a mouse model (spf-ash) facing acute metabolic decompensation due to PR8 influenza virus infection in Ornitine Transcarbamilase Deficiency (OTC-D). They studied the model in physiologic conditions and far from the effects of ammonia scavengers to highlight the direct effect that the virus has on urea cycle. Further studies would be needed to clarify the possible effects of drugs on the pathophisiology of the PR8 influenza virus on urea cycle.

In UCD  ureagensis is always compromised and infection places a greater burden on the cycle as protein breakdown is acclerated

R.Dear reviewer we have now inserted this concept in the paragraph, thank you.

The work by Tarasenko is not explained in any detail how do amino acid disturbances in UCD due to infection extend beyond the liver? What about more evidence

R.Given the need of reducing the lengh of the paragraph, and in the light of not enough clear and focused concept, we decided to move away this last sentence from the paragraph.

Fatty acid oxidation MCAD is not a severe form of FAO- of all the FAO defects this is the one only needing treatment in illness and can be treated by a normal diet and an emergency regimen in illness. After the age of 12m a 12 h fast is recommended- unlike the other FAO which commonly need overnight feeding and TFP is known to have poor immuninology.

R.About MCAD, by including it in the most severe forms of FAOD, we meant to consider the most genetically severe forms, requiring pharmacologcal treatment and short time fasting. However, we agree that it is rather unclear and we moved it away.

CD8T cells for the reader it would be good to have some background about these cells e.g lympocytes with the ability to kill virus infected cells. I would argue that in any IMD (FAO/ UCD) any stress due to an infection leads to abnormal metabolities as the pathway is compromised. My struggle was how does this relate to the title not just immune function?

R.Many thanks for giving us the possibility to clarify. We have now specified the role of CD8T cells, reporting the reference. We absolutely agree that a stress factor able to determine acute metabolic decompensation, is always associated to altered metabolites, the title referred to the pathogenetic mechanisms beyond the altered metabolites, anyway we moved it away.

Diet and altered immune function there is no greater explanation on the sentence 'indeed an increased susceptibility to infections due to  decreased immunoglobulin levels has been reported in PKU patients but this is only referenced by one paper and no explanation given to how. Additionally PKU subjects are rarely admitted to hospital with infections compared to UCD/FAO this needs to be addressed

R.We have now clearly stated that the evidence come from a single study, and that explanation behind the observation of lower than normal immunoglobulin levels, were not provided. Also the lower prevalence of hospitalization of PKU patients due to infection compared to other IMDs, has been stated.

Groppers paper is valid but what about more recent findings? I think the findings on Zn and iron could be challenged this is just one reference

R.Other references reporting oligoelement status in PKU patients have now been added, specifying the importance of micronutrient monitoring also to prevent suboptimal immune system functioning.

Could it be argued that in mitrochondrial disorders immune resonses are chaged due to the defect in the mitrochondria, central to energy production? Which is the interplay between infections and IMD

R.Mitochondrial disorders represent the paradigm of the relationship between altered energy production and immune response alteration. This is clearly stated in the paragraph with the words: “MD patients have similar presentations to patients with primary immunodeficiencies, including unusual infections not commonly diagnosed in immunocompetent people” and in Table 1, where many examples of mithocondrial disorders with related references are reported. In this case, it is the altered metabolic pathway (energy production) to be the cause of infection and not the contrary (Figure 1).

Table 1 need the names of ALG 12-CDG/ MOGS

R.We have now reported the full names of ALG12, MOGS and PMG3-CDG in Table 1 and in the text.

Glycogen storage disease - what about the new findings Empaglifozin

R.Many thanks for this comment, which allows us to insert information also on this promising therapeutic approach to neutropenia in GSD1b patients. Now information and relative references have been added.

Arginine/ citrulline are supplemented in most UCDs

R.We have better clarified this concept in the text: arginine or citrulline, already used in the clinical practice , are particularly beneficial for UCD patients during infection”

In OA mention unusual bacterai but what type and what it the mechanism of action

R.The unusual bacterial infections have been mentioned as well as the possible mechanisms behind.

What about gut microbiota and immunity?

R.We have now inserted information and references related to this interesting relationship, many thanks for pointing out this topic.

Other findings

Line 37 typo and death if not properly and rapidly treated

R: OK, modified

Line 40 repeat word acute

R.Word “acute” deleted

Lack of references for lines 46 to 50 only one reference. Ref for energy use for immune systems, only one reference for metabolic disorders and immune link

R.We have added 5 references for this topic, thanks

Line 53 spelling Lymophcytes

R.Word amended.

Line 57 to 59 spelling predisposing not predisponing

R.Word amended.

Spelling OTC  ornithine transcarbamylase deficiency most common but no reference to how common

R.We have inserted the range of incidence, according to one of the most updated reports: Lichter-Konecki et al GeneReviews® [Internet]. Seattle (WA): University of Washington, Seattle; 1993.2013 Aug 29 [updated 2022 May 26].

Line 100 deputated? FAO energy is produced continually for movement not just in resting or stress

R.We agree, the sentence is umbiguous, so we have rewritten it in: “…particularly involved in the energy production during fasting and stress episodes

Line 142 is difficult to read hydoxylase sentence structure needs re wording

R.Thanks for that. We have rewritten the sentence: “Phenylalanine Hydroxylase (PAH), which converts Phenylalanine (Phe) into Tyrosine (Tyr)”

Line 143 English  since birth should be from birth

R.Word amended

Line 162 ? spelling and meaning subtend corrispective ?

R.Sentence rewritten in: “two different mechanisms, underlying corresponding IMDs”

Line 178 does not make sense

R.The sentence is related to pattern recognition receptors (PRR) signaling, we have now rewritten and added their function.

I think more work is needed on this paper for it to fully match its title, more references and evidence and arguments presented. There is a lot of work gone into the manuscript but in its current form I do not think it fully answers the question you have proposed.

R.This review reports the current available evidences on the mechanisms behing the relationship infections/IMDs. We have now risponded to all the reviewers comments and added more references (now 103), we do believe that the paper has much improved. We agree with the reviewer that probably, at this state of art, it is not possible to completely answer the question reported in the title. For this reason, we prefer to change it. moving away the second part.

Reviewer 2 Report

ID: microorganisms-2577364

The reciprocal interplay between infections and Inherited Metabolic Disorders: not just immune dysfunction. by Tummolo A and Melpignano L.

To the Authors:

General comments:

The authors reviewed the mechanism of the relationship between inherited metabolic disorders (IMDs) and infections.  It was considered that this review is described well and summarizes essential points; however, several points should be addressed to improve the manuscript.

Specific comments:

1. As the authors mentioned in the manuscript, the metabolic disruption may be directly linked to the dysfunction of the immune system in IMDs (for example, in congenital glycosylation disorders).  Please add this point in the Figure 1 and tune up the Fig. 1 with coloring.

2. The authors concluded that the relationship between infections and IMDs is important for the better management of IMDs.  Please provide some clinical suggestions for the better management of IMDs regarding infectious conditions.

3. Please refer to the other major factors that may be associated with the immune system in IMDs, including endocrine dysfunction.

4. It would be better to include the metabolic conditions related to aging and gender differences in terms of the points of infection.

Author Response

General comments:

The authors reviewed the mechanism of the relationship between inherited metabolic disorders (IMDs) and infections.  It was considered that this review is described well and summarizes essential points; however, several points should be addressed to improve the manuscript.

Specific comments:

As the authors mentioned in the manuscript, the metabolic disruption may be directly linked to the dysfunction of the immune system in IMDs (for example, in congenital glycosylation disorders).  Please add this point in the Figure 1 and tune up the Fig. 1 with coloring.

R.Thank you for this comment. The direct impact on immune system of some IMDs like CDGs, is already reported in Figure 1, with the specified mechanism “Altered receptor-ligand recognition”. We have toned-up the coloring, to make the different parts more evident.

The authors concluded that the relationship between infections and IMDs is important for the better management of IMDs.  Please provide some clinical suggestions for the better management of IMDs regarding infectious conditions.

R.We have now provided some examples of clinical management of infections in the setting of IMDs, also including some examples of immune dysfuntion-targeted interventions for CDGs and mitochondrial disorders.

Please refer to the other major factors that may be associated with the immune system in IMDs, including endocrine dysfunction.

R.Thanks for this observations. We have added this informationin in the discussion. We referred to the frequent observation of overweight and metabolic syndrome in IMDs, particularly in PKU adult patients, forecasting a possible adjunctive mechanism of altered immune system in these patients.

It would be better to include the metabolic conditions related to aging and gender differences in terms of the points of infection.

R.To our knowledge, information about sex prevalence of infections in IMDs, are so far not available. However, we have included, in response to the previous point, the age-related onset of metabolic syndrome and overweight in adult PKU patients.

Round 2

Reviewer 2 Report

ID: microorganisms-2577364

The reciprocal interplay between infections and Inherited Metabolic Disorders: not just immune dysfunction. by Tummolo A and Melpignano L.

To the Authors:

General comments:

It is considered that the authors successfully revised the manuscript according to the comments.